# An empirical network study of the antimalarial supply chain in Ghana

Osman Adams[ID][1,2☯], Chia-Lin Wang[3☯], Edmund Chattoe-Brown[4], Heather Hamill[5], Katherine Hampshire[6‡], Simon Mariwah[2‡], Daniel Amoako-Sakyi[2], Fiifi Amoako Johnson[2‡], Graeme J. Ackland[ID][3*‡]

**1** Department of Geography Education, Faculty of Social Sciences Education, University of Education, Winneba, Central Region, Ghana, **2** University of Cape Coast, Cape Coast, Central Region, Ghana, **3** JCMB Kings Buildings, Edinburgh, United Kingdom, **4** School of Media Communication and Sociology, University of Leicester, Leicester, United Kingdom, **5** Department of Anthropology, Durham University, Durham, United Kingdom, **6** Department of Sociology, University of Oxford, Oxford, United Kingdom

☯ These authors contributed equally to this work.
‡ KH, SM, FAJ and GJA also contributed equally to this work.
* gjackland@ed.ac.uk

## Abstract

A survey of a sample of Ghanaian pharmacies was undertaken to trace the antimalarial drug supply chain. We sampled antimalarial drug outlets in 6 districts across Ghana and traced their immediate suppliers. A third level of the supply chain was obtained by visiting these intermediate suppliers and finding who supplied them. This proved sufficient to track the supply of antimalarial drugs to major manufacturers or importers. We then used techniques of network analysis to study features of the supply chain. By mapping the network to a real geography we demonstrate that the network has a hub and spoke structure, and is dominated by companies in Accra, with a secondary hub in Kumasi. Regional centres such as Tamale and Cape Coast are of lesser significance. We used a range of network measures to analyse the network data provided by the survey. Degree distribution analysis suggests that one company, which appears to deliver direct to many customers, has a particularly dominant position in the network. However, PageRank analysis identifies a different company as being more influential. This company uses a different supply model, via intermediaries, which means that it has fewer direct links to retailers. Mathematical analysis reveals that the distribution network (defined by in-degree distribution over nodes) is scale free (Pareto-type), a characteristic of an unregulated free market for sellers. By contrast, the purchasing network (defined by out-degree distribution) appears to be more log-normal, showing limited agency for individual buyers. It is interesting that a single mathematical measure can capture the different challenges faced by sellers and buyers. Understanding the structure of Ghana's private antimalarial supply chain provides crucial insights for strengthening medicine distribution systems in other low and middle income countries. Mapping such networks can inform global strategies to improve equitable access and ensure medicine quality.

**Data availability statement:** All data used in this paper is contained within the manuscript or within the provided spreadsheet and codes at https://git.ecdf.ed.ac.uk/gja/ghananetworks. This is sufficient to reproduce the analysis and results presented. Personal identifying data of survey participants, irrelevant to the present analysis, were collected during the survey. These data were stored securely and are password protected to ensure confidentiality following the ethics review. Data are available from the University of Cape Coast Institutional Review Board for researchers who meet the criteria for access to confidential data. University Post Office, University of Cape Coast, Cape Coast, Ghana.

**Funding:** We received MRC funding for the STREAMS collaboration with grant MR/T022132/1. 630 The funders had no role in study design, data collection and analysis, decision to 631 publish, or preparation of the manuscript.

**Competing interests:** The authors have declared that no competing interests exist.

## Introduction

Ghana ranks among the 15 countries with the highest malaria burden in the world [1]. Malaria poses significant health challenges in the country, especially in rural areas where access to antimalarial drugs may be limited [2]. By analyzing the pharmaceutical supply chain network, the structure and dynamics of antimalarial drug supply can be better understood. This can aid in identifying areas with healthcare vulnerabilities, with the potential for enhancing healthcare services and infrastructure.

Frequent stock-outs of medicines in public-sector health facilities in Ghana mean that medicine supply is dominated by private retailers. These include over-the-counter medicine retailers [OTCs] and licensed pharmacies on which the majority of the population depend for antimalarial drugs [3]. These private retailers are supplied by wholesalers/suppliers, who control the supply network in terms of the quantity, flow and pricing of pharmaceutical products [4,5]. Empirical evidence suggests that this network of supply chains is highly vulnerable to penetration by substandard and falsified (SF) medicines, with an estimated 35% of artemisinin-based antimalarial drugs sampled in Ghana found to be substandard in a study in 2016 [6]. Like many Low- and Middle-Income countries (LMICs), regulatory capacity and enforcement is limited, both at the point of entry for imported medicines, and within the country via inspection of local manufacturing facilities and post-market surveillance [7–9]. It is therefore reasonable to assume that the private supply chains provide a good example of a self-organising network with no (effective) central control.

Supply chains comprise flows of products and payments between heterogeneous organisations including factories, wholesalers and retailers. The idea that these can usefully be conceptualised as social networks is far from new [10,11]. We can distinguish between managed supply chains within a large organisation, and self-organised supply chains which have no centrally-imposed structure. In a recent survey, MacCarthy et al. [12] have drawn attention to the fact that the practicalities of self-organised supply chain mapping are challenging and as a result empirical study of them is largely neglected.

This article reports a project to map private sector supply chains for selected anti-malarial drugs in Ghana. The article also reports preliminary attempts to characterise the resulting supply networks and discuss their implications. This work was undertaken as part of a wider project exploring the ethnographic rationale of supply chains (why trade occurs between specific organisations) and – ultimately – what bearing supply chain structure and operating mechanisms might have on the problem of substandard and falsified medicines (SFM). These are medicines that are less effective than they should be because, for example, they spend too long in transit and their active ingredients lose efficacy, or because they are illegally counterfeited with no or low levels of active ingredients, and sold as if they are genuine. Counterfeit products are often virtually indistinguishable from real medicine without using sophisticated methods of authentication which are unavailable to regular customers [13,14]. The impact of such SFMs is very serious, both in terms of avoidable deaths and, potentially, increased drug resistance in infective organisms that are not completely eliminated during treatment.

To fully understand the nature of the supply chain networks, we conducted an extensive survey of a sample of OTCs, pharmacies and suppliers in Ghana. This enabled us to make a detailed map of a significant part of the national supply network, including the quality of the drugs reaching consumers, and the efficiency of the supply chains. In this paper, we show how network measures can help us to understand the network. The information can serve as the foundation for surveillance and tracking of the nationwide distribution of antimalarial drugs. It also has the potential to enhance resilience to disruptive events (such as pandemics or natural disasters) and support better recovery from such events. The patterns observed in Ghana are not unique. Across sub-Saharan Africa and many LMICs, private supply chains dominate the distribution of essential medicines. Mapping these systems as networks can help global policy makers identify vulnerabilities. Since there are many thousands of outlets selling antimalarial drugs in Ghana, it was necessary to conduct this research based on a sample, designed to represent the many diverse socioeconomic and agroecological regions across the country. It seems that to some extent supply chain mapping as a social network phenomenon falls between two different conceptualisations. The great majority of supply chain research considers cases where an organisation has an overview of the system and therefore mapping is trivial. There is no need for a company to establish where it has its own factories and warehouses. By contrast, even enumerating outlets participating in private sector pharmaceutical supply chains requires significant research effort. Some of these premises are licensed and therefore can theoretically be accessed via a government database; in practice, however, the database is incomplete, while other outlets are unlicensed. For these reasons, effective sampling has to take place on the ground through fieldwork. Here we specifically had to exclude, on practical grounds, peripatetic sources of pharmaceuticals such as irregular market stalls and travelling pedlars even though there is considerable concern about the role of such transient sellers in distributing SFMs [15]. Social network analysis tends to emphasise data that is in some sense manageably bounded (e.g., members of an organisation) or can be gathered automatically online. Neither conceptualisation applies particularly well to geographically distributed private sector supply chains. For example, it is not clear which standard social network measures apply effectively to sampled subnetworks, although some consideration has been given to measure robustness for missing data [16].

In this case, even the lower bound provided by sampling only licensed outlets gives a very large number of outlets that potentially need to be covered by fieldwork. The commercial nature of the relationships involved and local conditions mean that surveys would almost certainly be unsuccessful (or would only succeed in identifying disconnected fragments of supply chains, which defeats the point of a network analysis). Our strategy was therefore to start by sampling outlets dealing directly with customers, and then follow up the nominations they made for suppliers, repeating this process until we reached an importer or manufacturer. These random samples were constructed in different regions in order both to understand network structures across a wider area of the country given limited resource, and to investigate subsequently whether there were differences in the supply chain that might be explained by other relevant regional variations, such as road quality, settlement and population density, socioeconomic differences, and so on.

Unlike many social network studies, physical locations are essential to the nature of a supply chain network. These determine transportation times (and therefore potentially the degradation of medicines), the feasibility of inspection for SFMs (far greater near large settlements since travel can be challenging and time consuming) and perhaps even the likelihood of counterfeits being introduced into the supply chain (though finding out much directly about this kind of criminal activity is both very difficult and risky). All sampled outlets therefore had their location coordinates recorded in a form suitable for use with a Geographic Information System.

The empirical status of research in a particular interdisciplinary area is one of the least suitable topics for reviewing the literature. Searching for terms like "pharmaceutical supply chain" tells one nothing about whether the resulting studies involve data or not and there are no really discerning search terms to decide whether particular studies are empirical *a priori*. We have therefore attempted the best heterogeneous search strategy that we were able to devise to justify our claim that this study is the first empirical mapping of a geographically situated pharmaceutical supply chain. Firstly, a recent review [12] specifically looking for empirical mapping studies, discusses no cases for "pharmaceutical", "drug" or

"medicine" and following up its references brings none to light. The same absence can be seen in another recent and relevant literature review by Bento *et al.* [17] on applications and metrics of Social Network Analysis for supply chain management. Neither does looking at any subsequent study that cites these two review articles reveal any empirical work outside single organisations. On Web of Science a whole range of search terms anywhere in the articles being searched yield no hits at all. A search of Social Networks and Network Science, probably the two most important specialist journals in this area, for "supply chain" or "supply network" yielded 26 hits, none of which displayed an empirical mapping and many of which dealt with domains irrelevant for our purposes like illegal drugs. Interestingly, to support the point made earlier about the preference for bounded domains, two studies that were able to map supply chains, albeit in the unrelated area of farming, did so by piggybacking on data already collected officially for animal health tracing.

We thus conclude, provisionally but with some confidence, that this article reports the first mapping of a pharmaceutical network including geographical properties.

## Materials and methods

### Study context: Ghana

Ghana is a West African country, situated on the coast, and has long land borders with Togo, Burkina Faso and Côte d'Ivoire. Ghana's population is estimated at 30.8 million [18]. The country has 16 administrative regions across three main agroecological zones (southern coastal, central forest belt, and northern savannah) (Fig 1). The agroecological zones influence the socio-economic characteristics and distribution of malarial prevalence. Coastal and forest belt have lower prevalence of multidimensional poverty (income, energy, food, formal education, health) compared to the northern / savannah zone [19,20]. In addition, despite its high prevalence of malaria, the savannah zone is seriously challenged with limited health facilities and health personnel and lack of access / affordability of health care for the household [5,21]. Similar patterns exist in rural areas irrespective of the ecological zone. As noted above, medicine distribution and sales are mainly privately owned, with limited quality control measures.

Ghana is the second-largest manufacturer of pharmaceuticals in West Africa (after Nigeria), with more than 38 companies producing 30% of the country's medicine supply [22]. The remaining 70% are imported mostly from India, China and a few European countries through the only legal ports of entry via the main airport in Accra (the capital, situated on the southern coast) and its neighbouring seaport, Tema. However, our previous work suggested that some quantity of medicines enters the country illegally across its land borders [6].

While Ghana provides a specific national context, its mixed public-private pharmaceutical structure and regulatory challenges are representative of broader issues in medicine distribution across low and middle-income countries.

### Survey data

The data reported in this paper derive from a survey conducted in 2021 as part of the 'STREAMS' project (Strengthening private-sector Medicine Systems to tackle the persistence of poor-quality medicines in Africa), funded by the UK Medical Research Council [23].

Data collection was undertaken in two distinct phases: a 'retailer phase' and a 'supplier phase'. In phase one, the research team visited 120 licensed retail outlets (pharmacies and over-the-counter medicine shops), sampled from six of the country's 16 regions [5]. The six districts were representative of the three main ecological zones in Ghana: Bawku Municipal Area and Central Gonja District (Savannah/Northern), Kumasi Metropolitan Area and Bia East District (Forest/Middle Belt), Accra Netropolitan Area and Ketu South Municipal Area (Coastal/Southern). The criteria for selecting these areas ensured that the sample would include border and non-border, and rural and urban districts. Differentiating border and non-border districts was because border areas are known to be points of high smuggling of drugs into the supply chains. A spread of urban and rural locations was chosen in order to reflect the differences in socio-economic

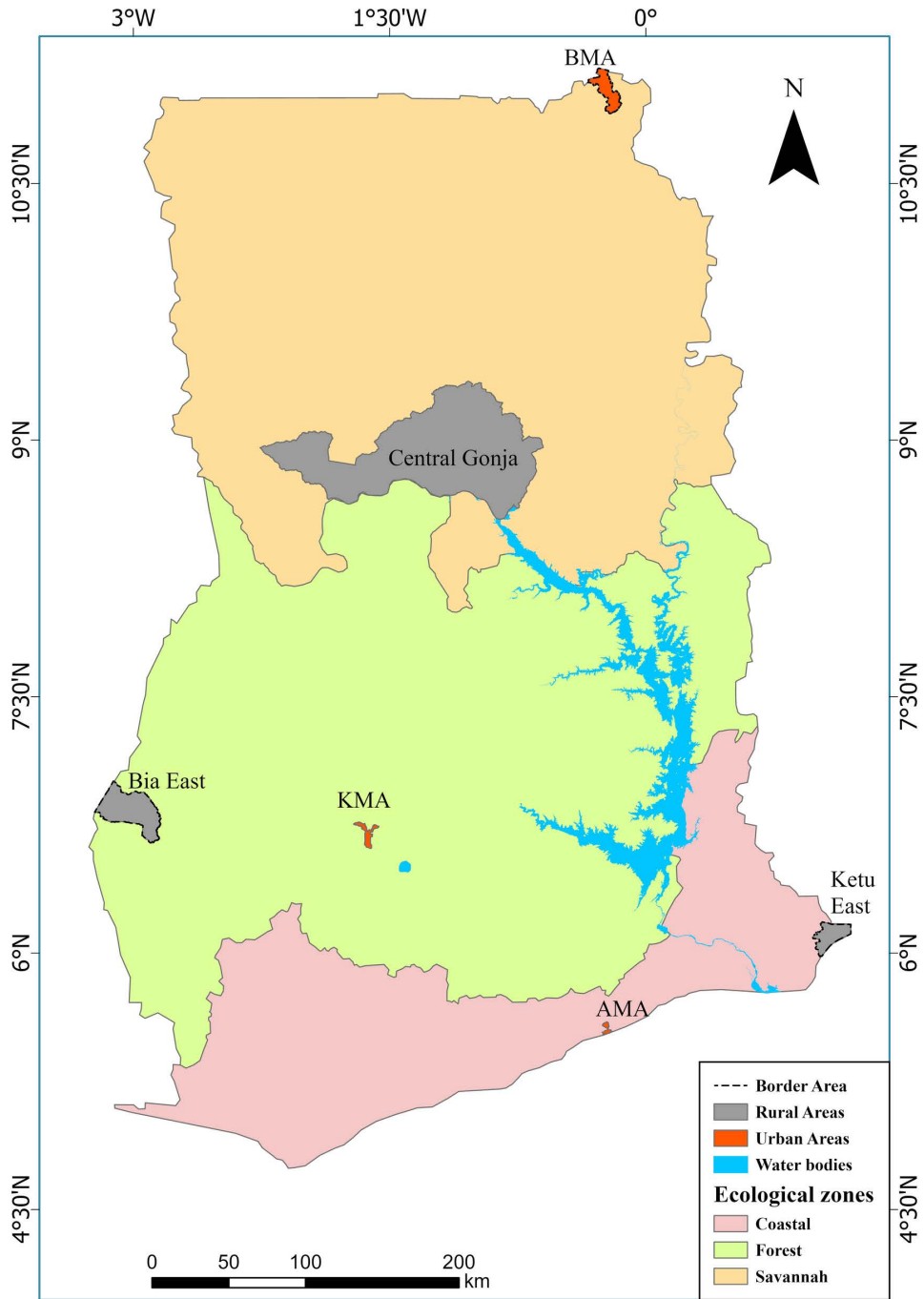

**Fig 1. The six study districts in the context of their agro-ecological zones and border/non-border and rural/urban characteristics.**

characteristics which influence access to health facilities. A total of 120 outlets were selected: 20 in each district (10 pharmacies and 10 OTCs). In areas where there are less than 10 pharmacies, all were selected and supplemented with additional OTCs [5]. The survey is illustrated in Fig 1.

Each of the 120 outlets was visited in person by a member of the research team, who administered a survey to collect data on antimalarials (artemisinin combination therapies and other medicines used to treat acute malaria in adults) currently in stock. For outlets with three or fewer such products, further data were collected on each one. For outlets with more than three antimalarials currently in stock, the retailer was asked to identify the three best-selling products in each of three price ranges: more expensive, intermediate, and cheaper. For each of those three products, the survey team captured data on the product itself (name, manufacturer, price, expiry date, etc.) and on the supplier of that product.

In phase two, the research team visited every supplier listed by retailers in the retailer survey In theory, this could have generated a sample of up to 360 suppliers (120 retailers listing the suppliers of each of three antimalarial products). However, since several companies supply more than one product and/or outlet, the total number of suppliers gathered from the survey with retailers was 69. Each of these suppliers was asked to provide information on the companies supplying them – this third level returned 40 "suppliers of suppliers." Some of these had already been identified by retailers in the first survey. Many of these 'suppliers of suppliers' were branches of major manufacturers or importers, indicating that three levels of survey is sufficient to obtain a thorough sampling of the supply chain in this case.

Participant recruitment took place between 15th July – 5th September 2021. All participants gave verbal informed consent, which was witnessed by another individual at the location (medicine outlet) and recorded by the research team. No minors were included in the research.

## Describing supply chains as formal networks

In our analysis, a pharmaceutical supply chain is conceptualized as a *directed network*. Network nodes represent the businesses, while links have a *direction* indicate the flow of drugs from manufacturer to customer. In the present case, end-user customers are not directly included in the supply chain analysis. The highest level of node are designated importers or (local) manufacturers. From these, out-links go to wholesale pharmacies, and thence onto licensed retail pharmacies and OTC outlets. Complexity arises from the fact that some businesses fulfill multiple roles, for example, serving both commercial customers (as wholesalers) and individual customers (as retailers). Moreover, from the survey, it is clear that connections might exist between all types of nodes: e.g., some importers deliver direct to some retail outlets.

To summarise, the four types of node are:

- Importer: out-links only
- Wholesaler: in-links and out-links
- Pharmacy/Wholesaler: in-links, out-links and customers.
- Licensed Pharmacy/OTC: in-links and customers.

## Directed network analysis

The network is fully represented by an appropriate Adjacency Matrix *G* (created using the `networkx` built-in function `nx.adjacency_matrix()`). Because the network is *directed*, this is an asymmetric matrix. The survey did not determine amount of medicine delivered along each connection, so the matrix comprises only 1s and 0s. The survey yielded a total of 229 nodes and 348 links, including 38 pharmacies, 82 OTCs, 69 visited suppliers and 40 further suppliers who were either not visited or declined to be surveyed. In many cases, the "suppliers" are also pharmacies. *G* is therefore a 229 × 229 matrix We represent the direction of each node as being towards the source of the drugs. To 'visualize' the adjacency matrix, we show the `seaborn` [24] heatmap for *G* in Fig 2.

Analysis of the network was carried out using python and its associated `networkx` and Folium libraries. A simple visualisation of the network is given in Fig 3. This shows the OTCs and pharmacies in the six surveyed regions, plus the

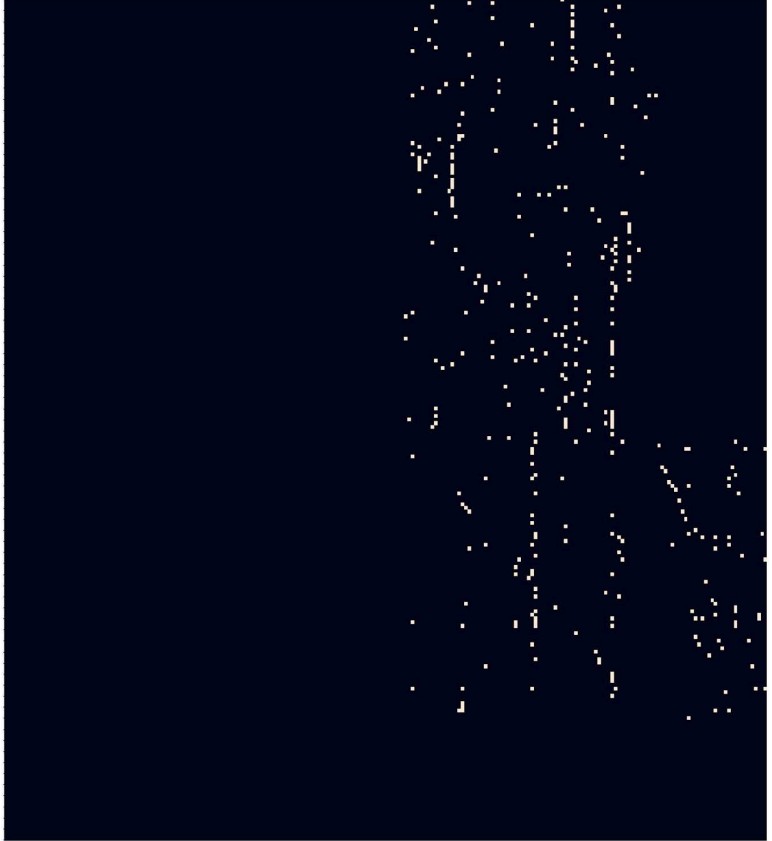

**Fig 2. Adjacency matrix of the supply chain network plotted as a "seaborn" heatmap: each node has a position along *x* and *y* axis, White dots indicate links (1's) from *x*-node to *y*-node.** Large black area on the left corresponds to zero in-degree for the pharmacies and OTCs (low node index), black area top right indicates no links from suppliers to retailers, and the black area at the bottom indicates no known out-degree for the top level suppliers.

suppliers spread throughout the country. One can immediately see the dominance of Accra, Ghana's capital and only legal port of entry for imported medicines, as a hub for the supply.

## Loop analysis

A supply chain is normally expected to have a tree structure with drugs flowing from the top level manufacturers through various nodes ending at the pharmacies. In such a network, there would be no loops, as we do not expect drugs to circulate back to suppliers. Mathematically, number of directed loops with n-links is given by the trace of $G^n$. For $n > 1$, this is indeed zero, however $\text{Tr}(G) = 6$: six self-loops ($n = 1$), are observed, shown as circles in Fig 3a. These self-loops indicate that the survey reports that a supplier is supplying itself. Considering that about 30% of Ghana's medicine supply is manufactured locally, and the self-loops are in highly-connected nodes, it is reasonable to assume that these self-loops indicate a business involved in both manufacture and supply from the same premises.

## Degree distribution

The degree of each node refers to the number of links associated with it. This can tell us, for example, whether a pharmacist has many possible sources or is reliant on a single supplier.

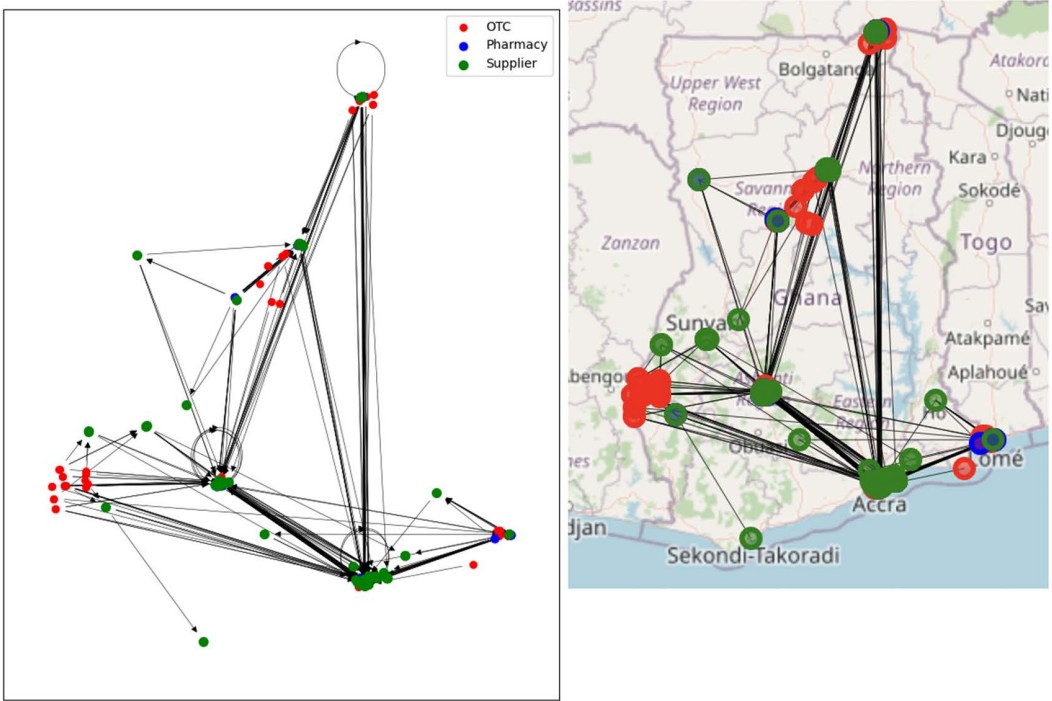

**Fig 3. The Ghana antimalarial drug supply chain network, illustrated as a `networkx` graph object and superposed on a map.** Node colours indicate the type of outlet; arrows are directed towards the supplier. The directions of links are only displayed in the `networkx` view.

The degree distribution is a probability distribution that indicates the probability ($p_k$) that a randomly selected node in the network has a degree $k$. Analyzing the degree distribution of a network identifies how many connections each node has [25,26].

For a directed network, one may independently consider the properties for both the in-degree (number of suppliers to a node) and out-degree (number of nodes supplied by a given node) distributions since they might exhibit different behaviours [25]. In order to identify the most appropriate network type for describing the antimalarial drug supply chain, three probability distribution models are applied to both the in-degree and out-degree distributions. An assessment of goodness of fit is then utilized to make this determination.

Fig 4 illustrates the degree distributions of the supply chain network, showing how diverse the range of available suppliers is. At first glance, one might intuitively anticipate the underlying distribution to follow a power-law or exponential decay for the in-degree distribution, and perhaps a log-normal pattern for the out-degree distribution. Thus these three models are fitted to the degree distributions using the function `scipy.optimize.curve_fit()`. The model functions are defined as follows [27]:

Power-Law fit: $p(k) = ak^{-b}$, $a, b$ are parameters to be estimated

Exponential fit: $p(k) = ae^{-bk}$, $a, b$ are parameters to be estimated

Log-normal fit: $p(k) = \frac{1}{k\sigma\sqrt{2\pi}} \exp(-\frac{(\ln(k)-\mu)^2}{2\sigma^2})$, where $\sigma, \mu$ are parameters to be estimated

The fitted curves and the corresponding optimized parameters are shown in Fig 4,

Upon simple visual inspection of Fig 4, it is apparent that the power-law fit is the most suitable for the in-degree distribution, while the log-normal fit best describes the out-degree distribution.

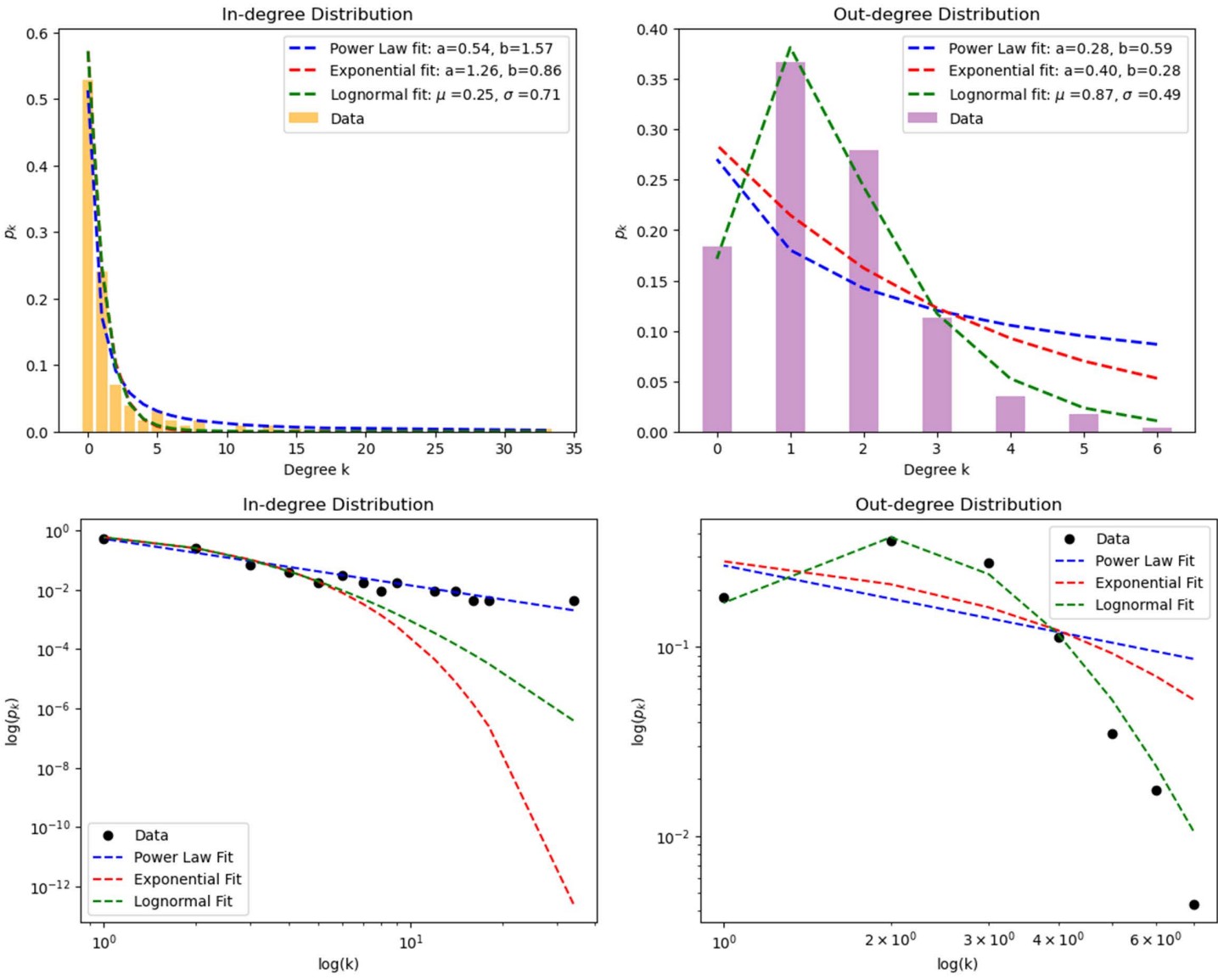

**Fig 4. Top: In-degree (left) and Out-degree(right) distributions of the supply chain network and their model fits.** Bottom: Data plotted in log-scale. (left) In-degree distribution, (right) out-degree distribution. Sample sizes are given in the open primary dataset (See Data Availability Statement for access).

To support this observation with statistical data, the $\chi^2$ and p-values for the three models compared to the original distribution data are calculated using the `scipy.stats.chisquare()` function.

For the in-degree distribution, with 13 degrees of freedom, the power-law fit yields the lowest $\chi^2$ value of 14.51 among the 3 models, indicating the least deviation from the original degree distribution, with a p-value of 0.3390 for rejection. In contrast, the $\chi^2$ values for the other two models are significantly larger, signifying a clear deviation from the data and leading to the rejection of their respective null hypotheses with >99% confidence.

On the other hand, for the out-degree distribution, with 6 degrees of freedom, the log-normal distribution yields the lowest $\chi^2$ value of 4.147 among the 3 models, with a p-value of 0.6567. Following the same approach, the hypothesis that the supply chain follows a log-normal distribution, cannot be rejected. However, the $\chi^2$ values for the power-law and exponential models are significantly larger, resulting in the rejection of their corresponding hypotheses.

Thus the $\chi^2$-test analysis of the fits supports the by-eye conclusion that the in-degree is a power law distribution, and the out-degree is log-normal.

## Consequence of scale-free properties

The power-law distribution indicates that a small number of major suppliers, such as Tobinco Pharmaceuticals (Tobinco Pharmaceuticals Limited is a leading pharmaceutical marketing and distribution company in Ghana, located in Accra. It has the largest in-degree in the network with $k = 33$.) and has a dominant position, supplying a multitude of retailers and small suppliers.

Fig 4 shows that the power-law fit for in-degree distribution has an optimized parameter $b = 1.57$. Since a power law $k^{-b}$ lacks a well-defined mean and variance with $b \leq 2$ [28], this implies that there is no typical value for the in-degree of a random node, hence the term 'scale-free' [25]. We can expect that the value of the largest in-degree node would be bigger if a larger survey was performed.

Scale-free networks are robust against random failures: Randomly removing nodes or links does not usually significantly impact the entire network. However, the network can be vulnerable to targeted attacks: for example, the failure of a highly connected node (such as a transport hub) can potentially disrupt the entire network [29]. For example, while the closure of a pharmacy in the Upper East region of Ghana is unlikely to significantly impact the drug supply in other regions, a distribution failure involving Tobinco could lead to a shortage of antimalarial drugs across the entire country.

We now consider the out-degree of the supply chain. The out-degree distribution in Fig 4 ranges only from 0 to 6. This indicates that most nodes maintain a moderate number of suppliers.

We note that the top-level 'suppliers of suppliers' were not visited, so those having an out-degree of 0 may be an artefact of the survey method. This could be a factor contributing to the observed log-normal pattern, however the very good fit of the data to this distribution compared with Normal or power-law suggests that this type of long tailed distribution is correct.

## PageRank

From the previous section, it was found that Tobinco Pharmaceuticals Accra (Node 183) has the highest in-degree, signifying the most links connecting to it. However, having the most links is only one measure of the most 'important' node in the network. This is because the assessment of importance requires consideration not only of the quantity of links but also whether a node is connected to other highly connected nodes. PageRank, an algorithm originally developed by Google to rank web pages in search results [30], assigns more importance to links from highly-connected neighbours. PageRank has been adapted for use in other directed networks as a method to assess and rank the importance of nodes in the network.

In mathematical form, the algorithm can be written as [30]

$$PR(A) = (1 - d) + d \left( \frac{PR(N_1)}{C(N_1)} + \ldots + \frac{PR(N_n)}{C(N_n)} \right)$$

(1)

In this equation, $PR(A)$ represents the PageRank of Node A, $d$ is a damping factor, typically set ot 0.85, and $N_1, \ldots, N_n$ are the nodes connecting to Node A. The terms $PR(N_1), \ldots, PR(N_n)$ correspond to their respective PageRank scores, and $C(N_1), \ldots, C(N_n)$ denote the number of outgoing links from nodes $N_1, \ldots, N_n$.

  

To understand this in the context of the supply chain network we imagine starting at a randomly-chosen pharmacy, and randomly choosing out links tracing back along the supply chain: the more often a supplier appears in such a process, the higher its 'PageRank'. To calculate and rank the PageRank scores for all nodes in the antimalarial drug supply chain network, the `networkx` built-in `nx.PageRank()` function is employed.

Surprisingly, the top ranked node in the supply chain, based on the calculated PageRank, is not the highest-connected node. Rather, PageRank identifies the East Cantonment Pharmacy (ECPL) in Accra as almost twice as highly ranked as Tobinco, which is the second most highly-ranked. Despite ECPL having significantly fewer links connecting to it, with an in-degree of only 11, it surpasses Tobinco due to the fact that the nodes connecting to it are 'better quality'.

To understand what this could mean in practice, consider two lorries loaded with antimalarial drugs, one departing from Tobinco Accra and another from ECPL. The Tobinco truck travels directly north, delivering its drugs to several small OTCs in the Upper East Region. On the other hand, the ECPL truck delivers to a single intermediate supplier, an example being PFA Pharmacy in Kumasi which in turn, supplies antimalarial drugs to 11 more retailers across various regions. Interestingly, this means that the antimalarials supplied by ECPL end up reaching more retailers and regions, despite having fewer direct linkages with suppliers compared with Tobinco. This phenomenon establishes ECPL as more 'influential' in the network.

Not only is the PageRank of ECPL almost twice the value of Tobinco, the next three highest ranked distributors have similar PageRank scores to Tobinco.

## Clustering coefficient

The *clustering coefficient* quantifies the degree to which the neighbouring nodes linked to a given node form connections among themselves. Mathematically, the *local clustering coefficient* of a node $u$ with degree $d_u$ in an undirected network is defined as

$$C_u = \frac{2L_u}{d_u(d_u - 1)} \tag{2}$$

where $L_u$ is the number of links found between the $d_u$ neighbors of node $u$ [25]. For a directed network, the definition is generalized to account for all possible directed triangles through node $u$:

$$C_u = \frac{T(u)}{2(deg^{tot}(u)(deg^{tot}(u) - 1) - 2deg^{\leftrightarrow}(u))} \tag{3}$$

where $T(u)$ is the number of directed triangles (Here, a directed triangle doesn't necessarily have to be a formation where three links originate from a node and traverse back to the same node. Instead, consider initially disregarding the direction and counting the triangles. Further details on how this expression is generalized can be found in the referenced paper [31].) through node $u$, $deg^{tot}(u)$ is the sum of in degree and out degree of $u$, and $deg^{\leftrightarrow}(u)$ is the reciprocal degree(number of links already pointing in both directions) of $u$.

## Sampling limitations

The supply chain data represent a partial sampled subgraph rather than a complete national network.

Retail outlets were drawn from only six of Ghana's sixteen regions and in equal numbers per district, introducing boundary specification effects and degree truncation at the retailer level. This was ameliorated by selecting the regions to maximise diversity.

The upstream sampling approach preferentially captures highly connected hubs. These features introduce degree bias and inflate apparent clustering around dominant suppliers [32] In particular, measures of centrality and influence—such

as PageRank—are likely to overstate the importance of well-sampled upstream firms and understate the role of sparsely connected or unobserved nodes, [33,34] in sampled data relative to complete population-level estimates.

## Ethics

The University of Cape Coast Institutional Review Board (UCC-IRB, Ghana) provided ethical approval for the survey work (undertaken 2020-21) entitled "Strengthening Private Sector Medicine Systems to Tackle the Prevalence of Poor Quality Anti-Infective Medicines in Africa." The ethical clearance number is UCCIRB/EXT/2022/21 from the UCC-IRB. The STREAMS project has Ethical Approval from the University of Durham reference ANTH-2020-07-07T17:25:33-dan0krh. Also, the sampled outlets provided informed consent to participate in the study. Outlets had the option to revoke their consent at any point throughout the interview process. The data were stored securely and password protected to ensure confidentiality.

## Results

### Analysis and comparison of sub-networks

A total of 10 distinct sub-networks have been established, comprising the whole supply chain network, three sub-networks categorized by medicine type, and six regional sub-networks. For each of these networks, eight network attributes have been calculated, as summarized in Table 1.

### Regional sub-networks

The survey was conducted by visiting six different regions in Ghana. To observe any significant differences in network attributes, we created a separate network for each region. This analysis aims to distinguish regions with potentially more efficient and direct supply routes from those that may experience longer supply routes characterized by multiple intermediate stops in the supply chain. The six regions include three rural areas (situated in Upper East Region, Volta Region, and Western North Region), and three urban areas: Kumasi, Accra, and Tamale. The survey data for each region were filtered based on the longitude and latitude of pharmacies and OTCs, and their supplier information was subsequently included. A `networkx` graph object was created for each region, producing 6 distinct regional sub-networks, Figs 5–10.

The regional network graphs show how drug supplies flow overwhelmingly from south towards north. In the network of the Upper East Region, Tamale and Kumasi stand out as two major stops in the supply chain; however, it is noteworthy that a significant quantity of antimalarials is also directly supplied from Accra. Conversely, no retailers in the south are supplied by regions to the north.

**Table 1. Network Attributes, results calculated for the whole network and 9 sub-networks.** $L_{max}$, $L_{min}$, and $L_{avg}$ denote the longest, shortest, and average supply chain lengths in the network, respectively. $C_{avg}$ represents the average clustering coefficient of the network.

| | Nodes | Links | Avg. Degree | $L_{max}$/ $L_{min}$ | $L_{avg}$ | Direct Links | $C_{avg}$ |
|---|---|---|---|---|---|---|---|
| Whole Network | 229 | 348 | 3.039 | 4/1 | 2.248 | 39 | 0.00902 |
| Upper East | 61 | 88 | 2.885 | 3/1 | 2.026 | 15 | 0.0144 |
| Tamale | 61 | 74 | 2.426 | 4/1 | 2.520 | 1 | 0 |
| Volta | 51 | 65 | 2.549 | 4/1 | 2.505 | 5 | 0.0217 |
| Accra | 53 | 67 | 2.528 | 4/1 | 2.204 | 5 | 0.00283 |
| Kumasi | 64 | 77 | 2.406 | 3/1 | 2.064 | 3 | 0.00346 |
| Western North | 67 | 87 | 2.597 | 4/1 | 2.063 | 9 | 0.0168 |
| Cheap | 201 | 230 | 2.289 | 4/1 | 2.245 | 22 | 0 |
| Intermediate | 186 | 134 | 1.441 | 3/1 | 1.442 | 90 | 0.00294 |
| Expensive | 165 | 82 | 0.994 | 2/1 | 1.012 | 80 | 0 |

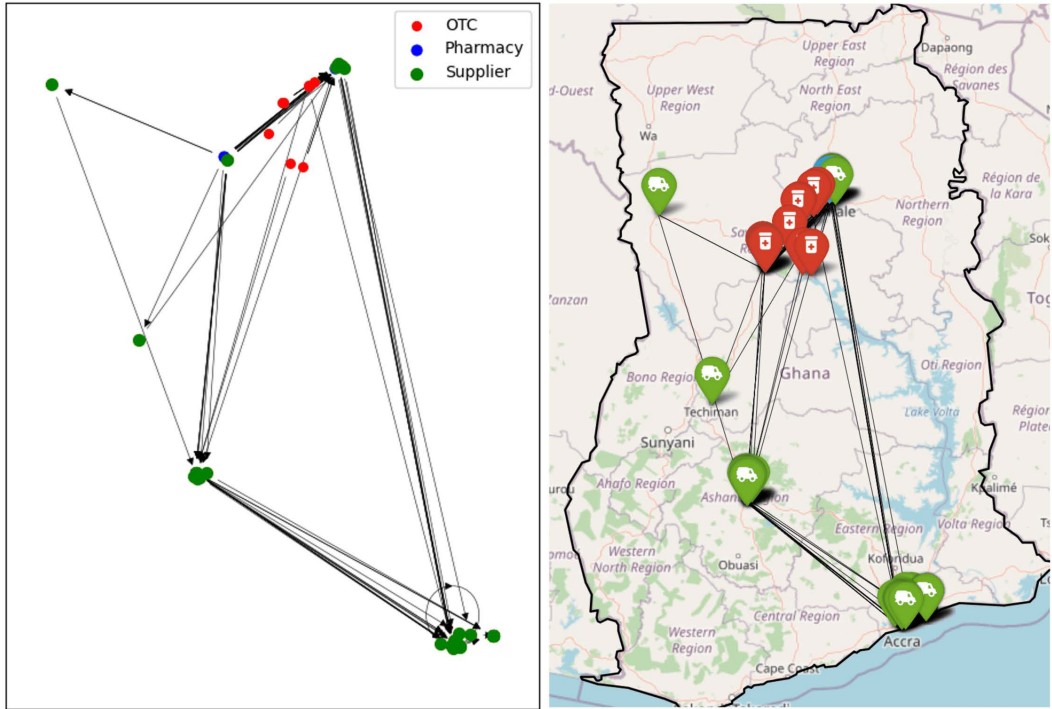

**Fig 5. Regional sub-network for Tamale shown schematically and on a map.** Node colours indicate the type of outlet, arrows are directed towards the supplier.

Given that Accra is the capital of Ghana and hosts the Port of Tema, the country's largest port, it is logical to assume that most antimalarial drugs are either imported to or manufactured in Accra. These drugs are then distributed throughout the country, a pattern clearly reflected in the regional sub-networks. Suppliers in the Accra Region have direct connections to every other region in the network.

One noteworthy observation is that almost all retail outlets (retail pharmacies and OTCs) in the Greater Accra Region sub-network have suppliers also located in Accra, with the exception of one supplier oddly situated in Kumasi (Shalina Laboratories Ltd Kumasi). Further examination reveals that this is a multinational company which has multiple branches across Ghana and is supplied by its factory in India. This anomaly may be attributed to internal stock distribution patterns within the company at the time of data collection. Nevertheless, in general, all retailers in Accra are closely supplied within the region. A more in-depth analysis, including an assessment of network properties, is discussed below.

## Sub-networks by medicine price category

As noted above, data were collected in the survey on the supply of antimalarial drugs, categorized into three categories based on their relative price as cheap, intermediate, and expensive. To explore potential variations in network structure, a different sub-network was created for each medicine "price category". The graph objects for each drug price category sub network are displayed in Fig 11.

No significant differences are observed in the networks for the different price categories, although the cheaper medicines have a slightly longer supply chain the more expensive medicine categories. The self loops are only observed in the cheap medicine category, supporting our hypothesis that these represent medicines manufactured locally.

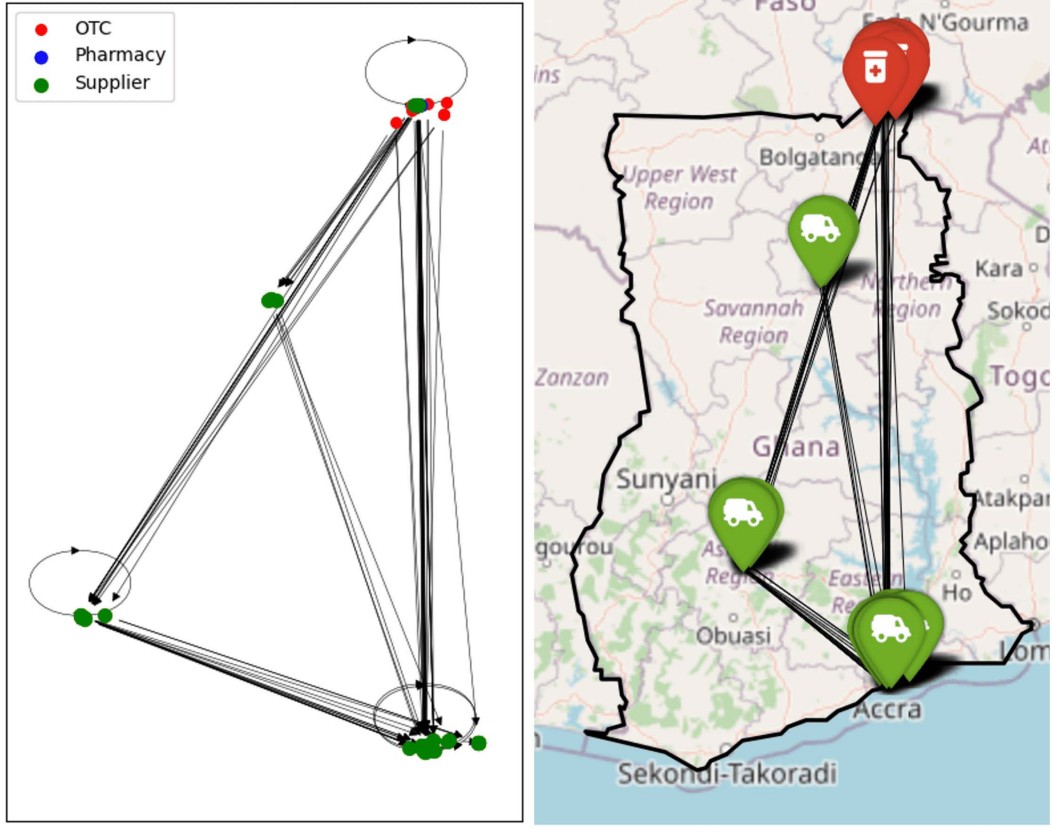

**Fig 6. Regional sub-network for Upper East shown schematically and on a map.** Node colours indicate the type of outlet, arrows are directed towards the supplier.

One potential flaw worth noting is that the classification of 'cheap', 'intermediate', and 'expensive' is relative rather than absolute during the survey [2]. This implies that a drug classified as expensive in one pharmacy, especially in a more rural area, might be classified as cheap in a more modern pharmacy in an urban area.

### Network attributes

**Path lengths.** A *path* is a route that runs between the nodes following the directed links. The *path length* between two nodes indicates the number of links on the specified path. There could be more than one path available between two nodes in the network, and the *shortest path* is the path containing fewest links between two nodes. The length of the shortest path is often referred to as the *distance* between them – not to be confused with geographical distance. The distance between the two furthest nodes (the longest shortest-path) in a network is known as the *diameter* of the graph. The distance and the number of shortest paths between two nodes can be calculated from powers of the adjacency matrix [25]. The elements of $(G^n)_{ij}$ are the number of paths of length $n$ between nodes $i$ and $j$.

Here, the important path length is the number of links a drug traverses before reaching the customer. This is defined by the number of links between intermediaries a retailer and its top level supplier in the network and excludes, e.g., links outwith Ghana to foreign manufacturers and exporters. The `networkx` built-in functions `nx.shortest_path_length()` and `nx.all_pairs_shortest_path()` are utilized to find the length of supply chain in the network.

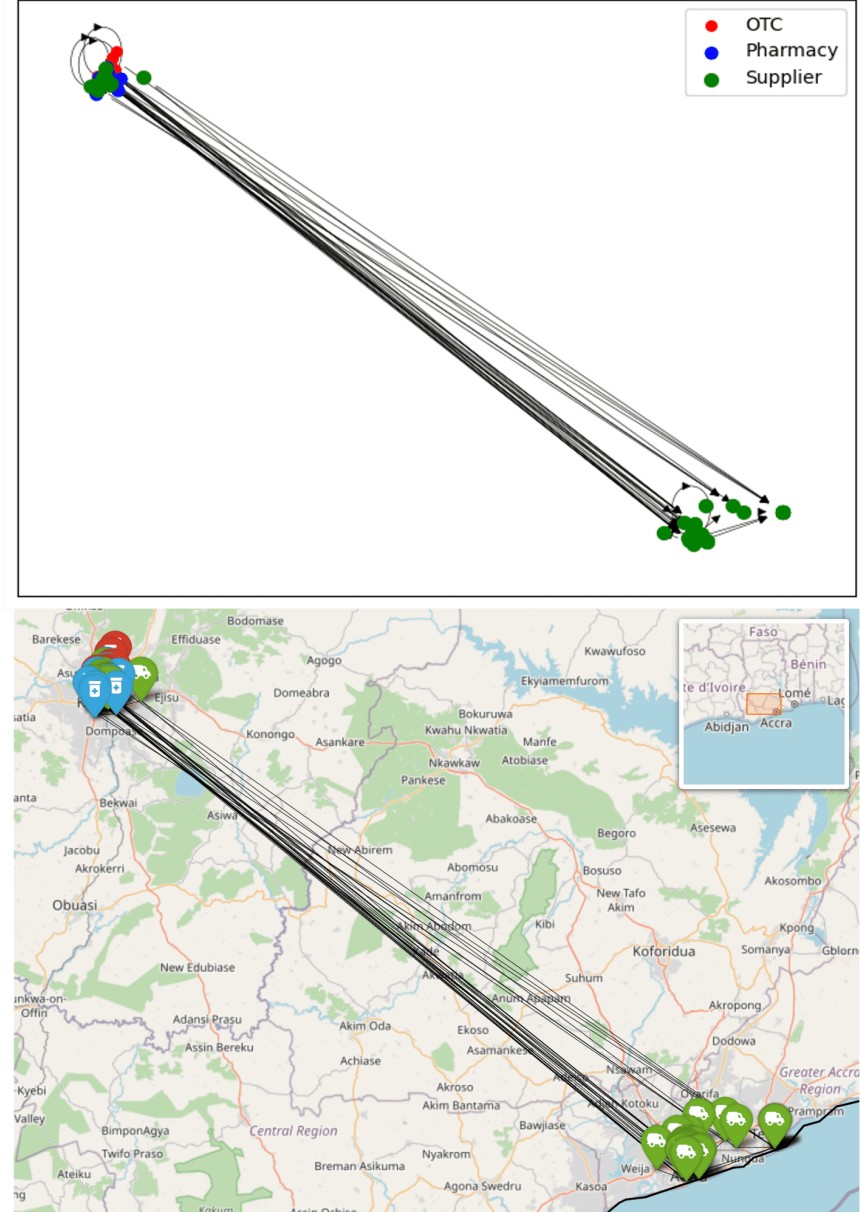

**Fig 7. Regional sub-network for Kumasi shown schematically and on a map.** Node colours indicate the type of outlet, arrows are directed towards the supplier.

In Table 1, $L_{max}$ represents the diameter of the network, indicating the longest supply chain length. $L_{min} = 1$ indicates that all sub-networks contain at least one direct link, meaning that drugs are supplied directly from a top-level supplier to some retailer in the network. The number of direct links found in a network is also recorded in Table 1. Additionally, $L_{avg}$ calculates the average supply chain length in the network:

Note that conventionally, the "average path length" of a network refers to the average of the shortest paths between **all** pairs of nodes in the network. However, given that we are dealing with a supply chain with a tree-like structure, we here

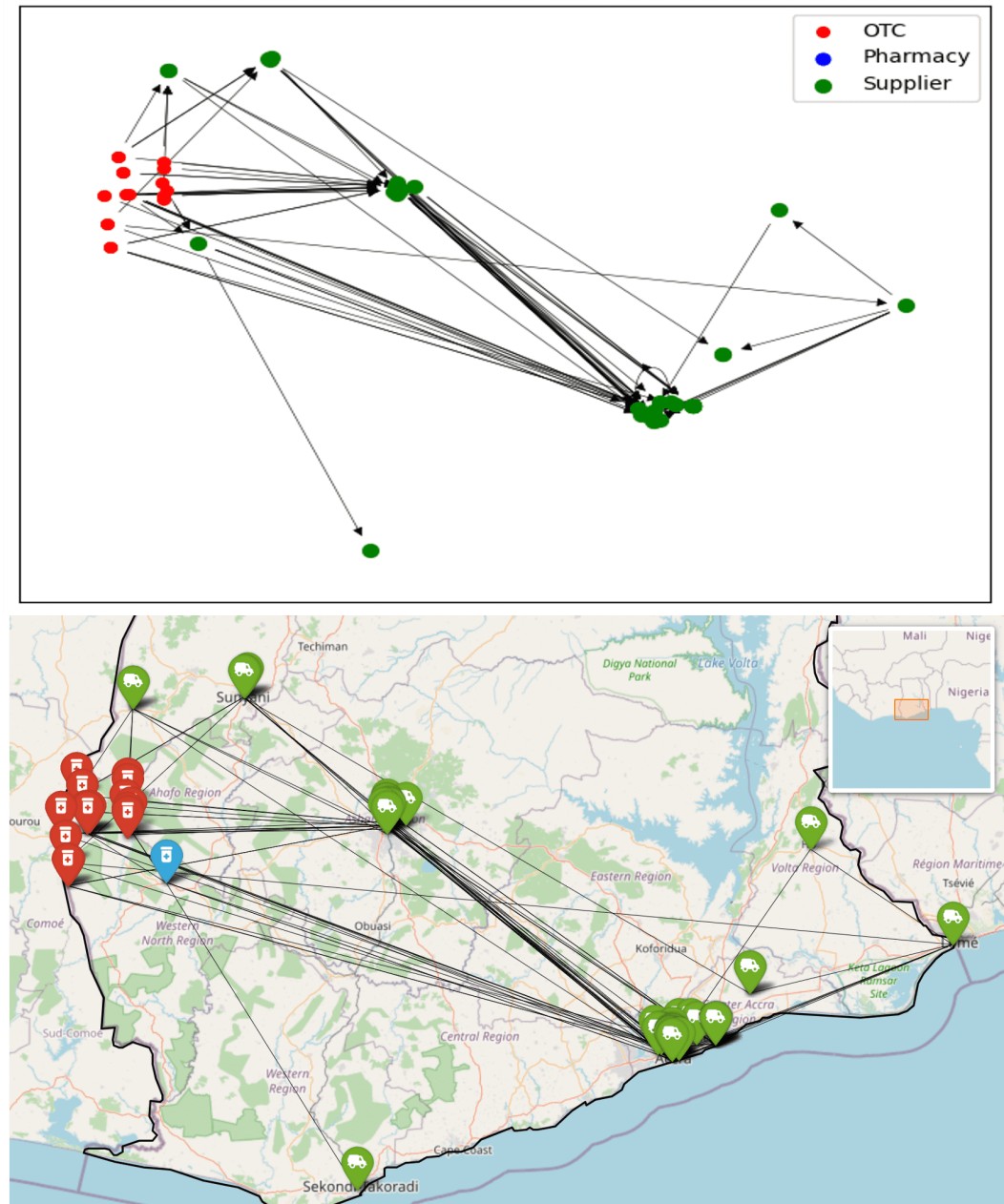

**Fig 8. Regional sub-network for Western North region shown schematically and on a map.** Node colours indicate the type of outlet, arrows are directed towards the supplier.

adopt a more meaningful definition of the (average) distance between a retailer and its top-level supplier. This reflects the average number of links a drug went through prior to reaching the customer.

For a convenient comparison of the numerical values Fig 12 presents bar plots illustrating information on nodes, links, direct links, and path length for each network.

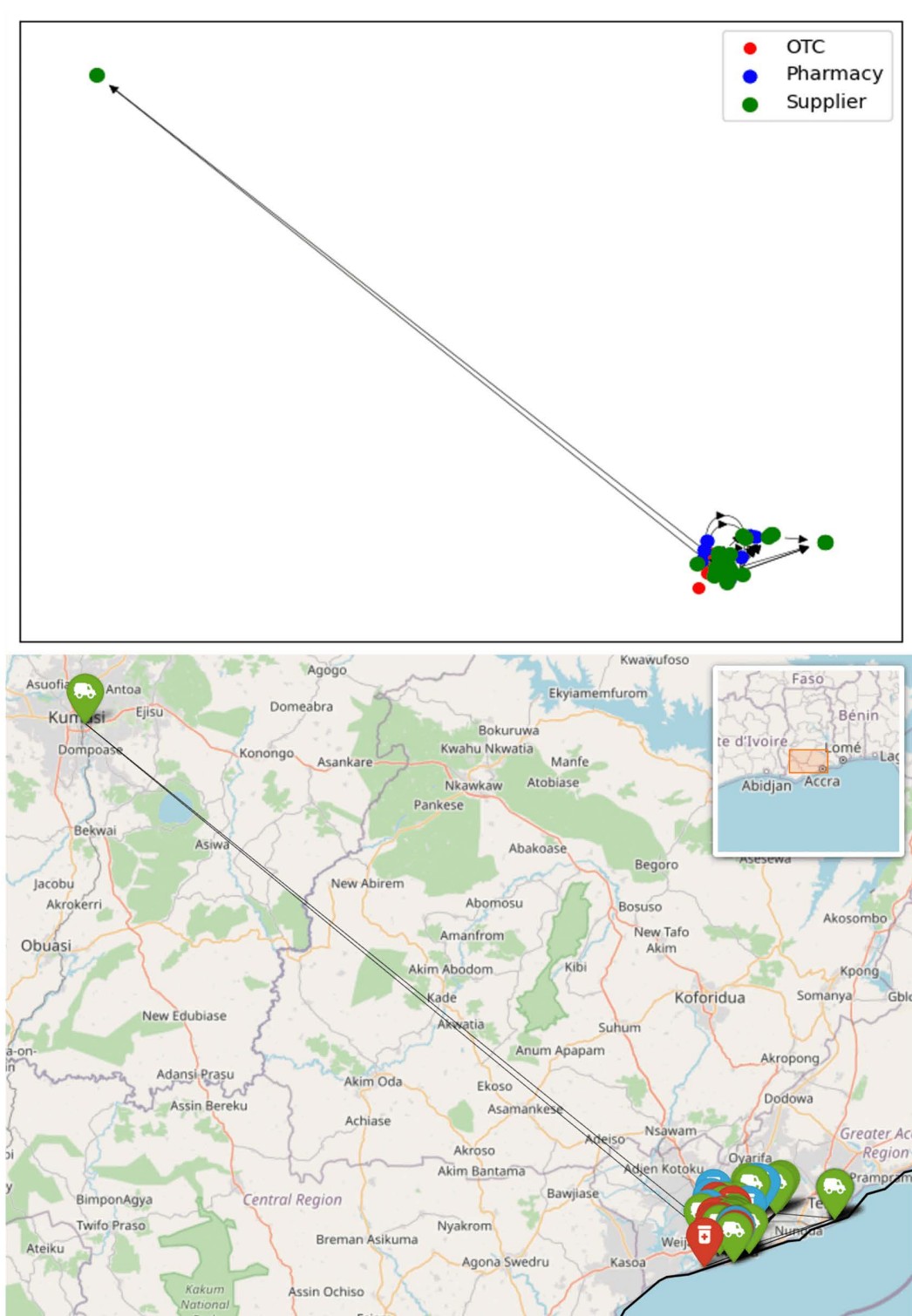

**Fig 9. Regional sub-network for Accra Region shown schematically and on a map.** Node colours indicate the type of outlet, arrows are directed towards the supplier.

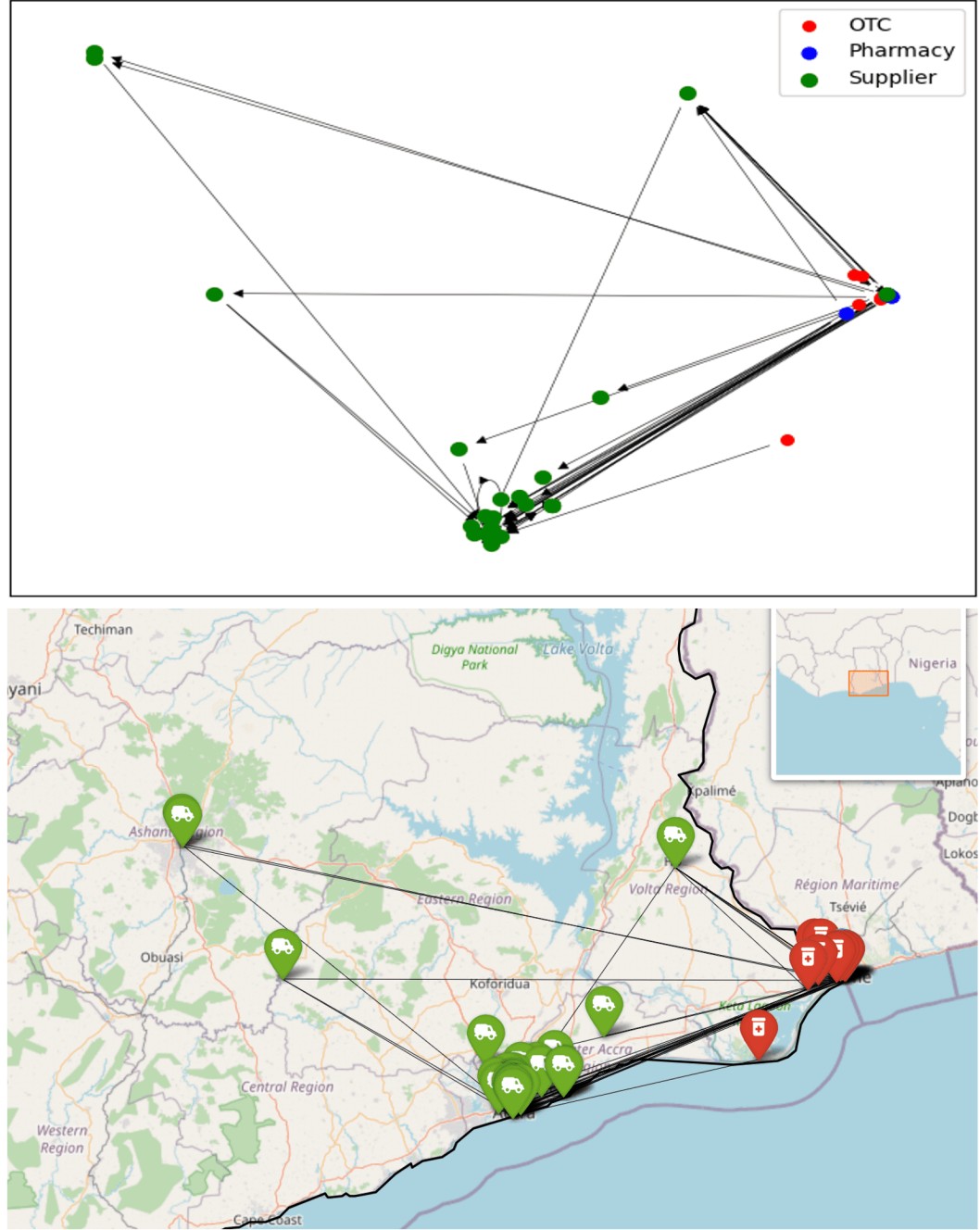

**Fig 10. Regional sub-network for Volta Region shown schematically and on a map.** Node colours indicate the type of outlet, arrows are directed towards the supplier.

Examining Table 1 and Fig 12 reveals a noteworthy finding for the Upper East Region. Despite its considerable distance from Accra, the region features 15 direct links between retailers and top-level suppliers, significantly surpassing the connectivity of other regional networks. Moreover, it maintains an average supply chain length of approximately 2.026 links, markedly the lowest among the six regions. This is interesting because one might expect that a greater physical

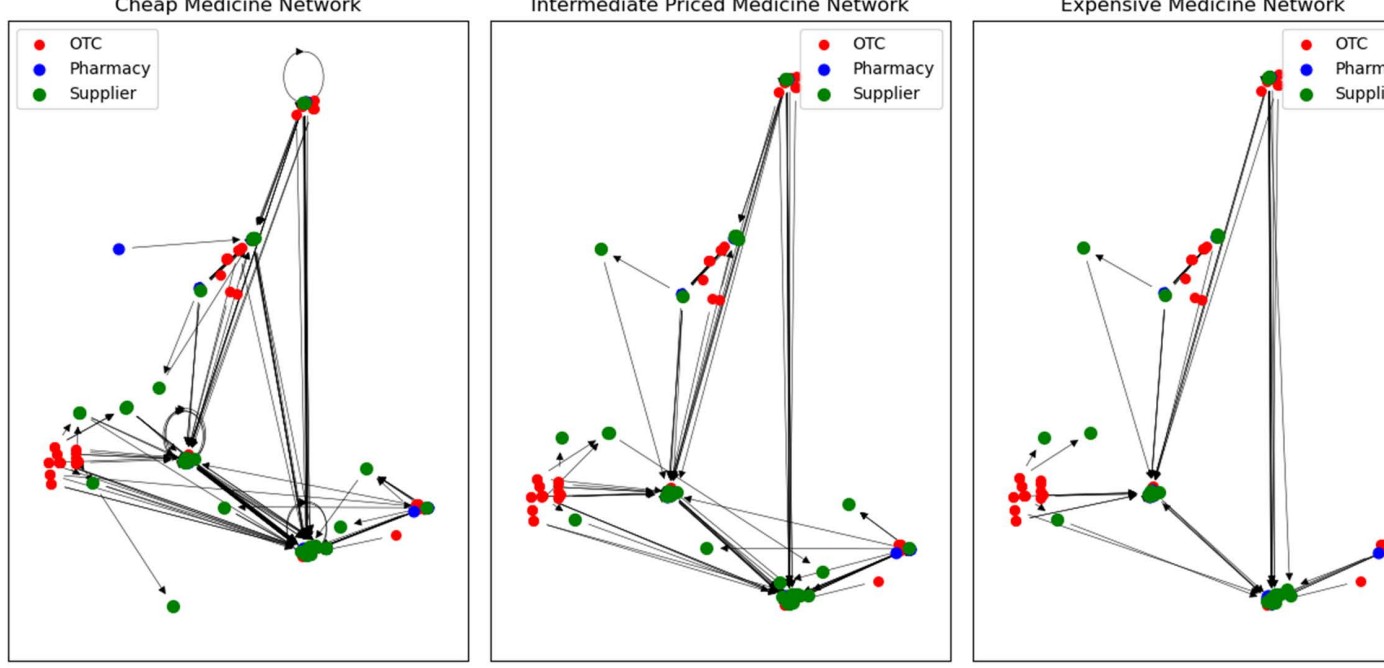

**Fig 11. Sub-networks by medicine price category, from left to right: cheap, intermediate and expensive.** Node colours indicate the type of outlet, arrows are directed towards the supplier.

distance implies more intermediary stops in the supply chain. However, the data implies that a truck could potentially cover the distance in a direct 14-hour journey from Accra to the region to deliver these antimalarial drugs. This shorter path-length may have implications for the timeliness of delivery and the quality of drugs supplied.

Another intriguing observation pertains to the larger number of direct links between retailers and top level suppliers within the intermediate-priced medicine sub-network and the expensive medicine sub-network. This can be attributed to the fact that higher-priced antimalarial drugs are less common in the market (indicating that fewer retailers stock them or fewer suppliers import or manufacture them), thus these drugs naturally pass through fewer links and nodes to reach the retailers. This is reflected by the reducing number of nodes and links in these two networks compared with the cheap medicine network. In the expensive medicine sub-network, of the 82 links present, 80 are direct links. This indicates that top-level suppliers are directly providing expensive antimalarial drugs to the selected retailers. For the remaining two links, there is only one intermediate stop between them, as suggested by $L_{max}=2$.

The lower availability of retailers stocking higher-priced antimalarial drugs could be attributed to financial considerations such as affordability or profitability. The relationship between price and quality for medicines is not known: although many interviewees in our previous work equated low price with poorer quality, others suggested that fraudsters might choose to copy high-end products, which might generate greater profit margins. However, if the price and quality of antimalarial drugs are positively associated, this could imply that higher-quality drugs are predominantly accessible in urban areas, leaving rural communities disadvantaged [2,8].

In Table 1, $C_{avg}$ shows the average clustering coefficient of the supply chain network. It is obtained by averaging through all the local clustering coefficients for each node in the network, calculated using Equations 2 or 3. The main feature of these numbers is that they are all very low, as one would expect for a tree-like network.

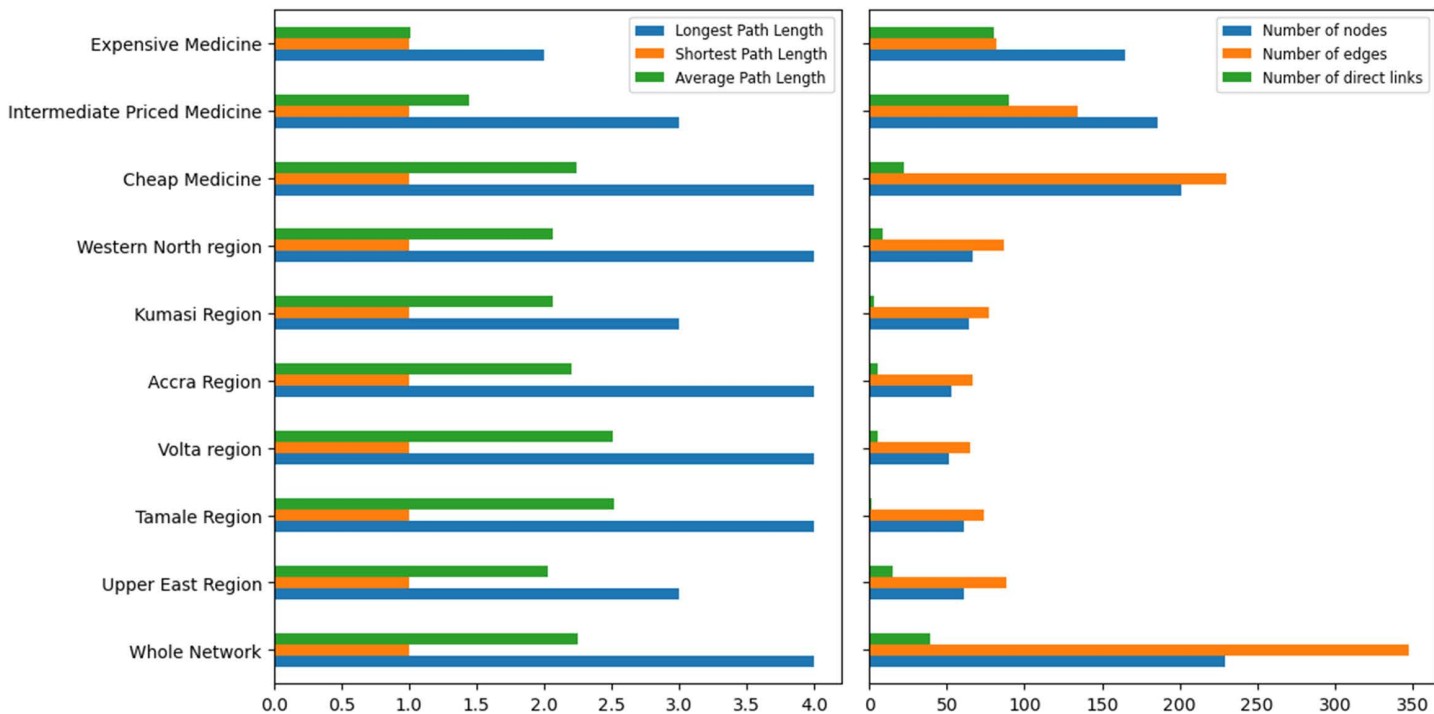

**Fig 12. (Left)** Longest, shortest and average supply chain path length, and **(Right)** Number of nodes, links and direct links in the whole network and 9 sub-networks.

Upon closer examination, it was identified that only three nodes in the intermediate-priced network had positive local clustering coefficient values. We know there are no directed loops in the networks, and it was found that all these cases are associated with situations wherein node A supplies B and C, and node B also supplies node C.

This scenario became meaningful in the sense that a top-level supplier supplies two other first-level suppliers, and one of these intermediate suppliers happens to also supply the same drug to the other. We can imagine this might happen if the pharmacy occasionally runs out of stock, it can quickly top-up from a nearby supplier rather than waiting for a delivery from the most distant top-level supplier.

Thus the clustering coefficient is a useful descriptor for a supply chain, in this case having a far lower value than one expected for a random network. Shorter and more direct supply chains may improve medicine quality and timeliness of access, suggesting that supply chain topology can have direct implications for public health outcomes.

## Discussion

### Weighted network

The survey did not report the quantities of medicines in each link. Consequently, we used an *unweighted network* for analysis, assuming that all links have the same weight in terms of the volume of drugs supplied along that link. In future work it could be useful to document the number of drugs and drug types being supplied. The introduction of a weighted network means that the the adjacency matrix would no longer consist solely of 0s and 1s. The mathematical descriptors of the

network could still be calculated straightforwardly, but this would increase the complexity in analyzing various attributes compared to an unweighted network [35].

## Dynamics

An intriguing topic for future research involves exploring the dynamics of the supply chain network. A *dynamic network* refers to a network that undergoes changes over time. In other words, the links within the network may shift or reconfigure over time, as suppliers expand or contract their supply areas; and nodes may emerge or disappear, as new outlets open and old ones close.

## Sample size

One significant limitation of this analysis is the small sample size. While there are approximately 4,600 registered pharmacies and around 21,000 OTCs in Ghana [36], only 120 retailers were surveyed for this project. The supply chain network derived from this limited dataset may not accurately represent the comprehensive structure of the antimalarial drug supply chain across Ghana.

Nevertheless, our survey is large enough to demonstrate that key descriptors of the network also hold true for smaller sub-networks. Based on this, it could be possible to build a representative network at the scale of the entire country. The key features of such a networks would be:

- close to zero clustering coefficient

- power law distribution of in-degree node distribution

- lognormal distribution of out-degree node distribution

- absence of directed loops

- 3–4 link supply chains

Similarly, the power-law in-degree distribution shows that the mean and variance of the in-degree links is not a meaningful measure, as it would increase as the network size increases.

To our knowledge, none of the standard algorithms for generating scale-free directed networks are able to simultaneously incorporate all these features: e.g., the Barabasi-Albert preferential attachment creates too many rings and an average path length which grows with system size.

## Sample structure

By design, the survey sample only included retail outlets in the six selected districts, and is therefore unlikely to be representative of the country as a whole. An indication of this would be if strong correlations were found between geographically-neighbouring outlets. For example, we found that the Upper East region has a relatively direct supply route from the same supplier in Accra. The same may not necessarily be true in the Upper West region (for example), and even if it is true, it may be a different company that dominates supply.

It would therefore be highly beneficial to expand the survey to include more regions and retailers. A relatively small survey in the Upper West region could quickly establish if outlets in this region also have a dominant local supplier. This expansion could also potentially enhance the accuracy of determining crucial suppliers in the country and aid in analysis of upper levels in the supply chain.

The partial network sampled by link tracing may fail to locate the least-connected suppliers, which can overestimate the tail of the in-degree distribution. This would imply that the value of the power, $b = 1.57$ is overestimated. However, we note that most suppliers are found more than once by the link tracing, which suggests that the number of unvisited suppliers is

low. Furthermore the fit in Fig 4 shows that even in the In-degree 1 and 2 suppliers are omitted from the data, the power law fit is robust.

## Impacting factors

The data analyzed in this project constitute only a portion of the information collected during the field study. In addition to details such as drug prices, supplier and location information, various other aspects of retailers were documented, including factors like the accessibility of outlets, the wealth index of the surrounding neighbourhood, distance to major healthcare facilities, and the overall conditions of the retailer stores etc.. Furthermore, detailed product information, including brand and manufacturer details, country of origin, and expiry dates, was also recorded. These factors could significantly impact the number of consumers visiting a specific outlet and influence their choice of antimalarial drugs. Such considerations might then have a cascading effect on the retailer's stocking decisions, subsequently impacting the supplier's choices in drug distribution [2].

Another interesting point is that not everyone seeks healthcare services in the event of illness. As well as personal preferences [37], factors such as perceptions of the role of pharmacists, reputation, trust, and personal beliefs can influence one's willingness to visit a particular drug retailer [8]. For instance, in Ghana, many patients prefer malaria treatment through locally produced herbal remedies over licensed pharmacies, citing reasons such as a preference for traditional practices [38–40] and anxieties about toxicity and side effects of pharmaceutical medicines.

These influencing factors are challenging to quantify and incorporate into the analysis of the general network structure of supply chains. However, they may raise questions about the conclusions drawn from pure data analysis. For example, an analysis might suggest that rural areas are at a disadvantage in terms of accessing high-quality drugs. Still, this observation could be influenced by relative poverty, or by the particular preference of people in rural regions, who may trust herbal products over licensed antimalarials or even traditional healers offering placebo treatments [41]. These dimensions are beyond the scope of this project, as they involve considerations that require delving into complex areas encompassing social, economic, and cultural aspects of healing and treatment seeking behaviours. Nevertheless, acknowledging these real-world issues will contribute to a more accurate reflection and may prompt a reconsideration of the results obtained solely through numerical manipulation.

From a public health perspective, understanding the topology of medicine supply networks provides a valuable diagnostic tool for policymakers. Identifying central nodes and weak links can inform interventions to improve resilience, reduce medicine stockouts, and prevent the spread of substandard and falsified medicines. This approach could help the WHO's Global Surveillance and Monitoring System on substandard and falsified medical products.

## Conclusion

Based on survey data, we have constructed and analysed the directed network of antimalarial drug supply chains in Ghana. This shows how network science can unlock some features of such a social network. The main result is to show how the supply chain network is similar to, but not exactly, a tree-like network. Rings do exist, either in the trivial sense of a drug produced also acting as an outlet, or as defined in Equation 3 to allow multiple pathways for drug to regions which might otherwise suffer stockouts. This latter feature is crucial for the resilience of the supply chain: a single point of failure in a tree leads to that entire branch losing supply, the ring provides an alternative supply route. Such rings are rare in this case, suggesting that in general the tree structure functions well.

Secondary features of the supply chain are the degree distribution, which is scale free for the in-links and log-normal for the outlinks. Also, the fact that the depth and path lengths seem to be independent of system size. This formal analysis can be cast into a narrative by envisioning the movement of lorries (think of them as the links) carrying antimalarial drugs. The highly connected but relatively low PageRank and short path-length of Tobinco indicates its

role in direct delivery to remote areas, while the high PageRank of other companies suggests a different model with a network of intermediate wholesalers. Further analysis of sub-networks, including regional sub-networks and medicine price categories, reveals that the overall flow of drugs is from the south towards the north of Ghana as one would expect given that manufacturing and import is concentrated around Accra. Moreover, the drug supply infrastructure in the Upper East Region is moderately directly linked to top-level suppliers. This reduces the direct supplier's overall influence as measured by PageRank, casting doubt on the usefulness of PageRank in this application. While clustering coefficient analysis provides insights into rare cases of drug supply issues, it is concluded to be less meaningful for the supply chain network. Finally, to extrapolate missing information from the network, a larger sample size or possibly an extension to a dynamic network is required. These points remain interesting areas for exploration in future research.

Beyond Ghana, this framework demonstrates how network science can contribute to global efforts to strengthen pharmaceutical supply chains in low and middle-income countries. As access to medicine and medicine quality are core pillars of universal health coverage, applying similar analyses everywhere could help ensure safer, more equitable, and more resilient healthcare systems worldwide.

## Acknowledgments

We thank Mark Naylor for help with the GIS mapping, and Kate Kilpatrick for proofreading and beautifying the text. We acknowledge OpenStreetMap for use of map data in Figs 1,3,5–9 under the Open Database License (ODbL) https://www.openstreetmap.org/copyright.

## Author contributions

**Conceptualization:** Graeme J. Ackland, Osman Adams, Heather Hamill, Simon Mariwah, Katherine Hampshire.

**Data curation:** Osman Adams.

**Formal analysis:** Graeme J. Ackland, Osman Adams, Edmund Chattoe-Brown, Chia-Lin Wang.

**Funding acquisition:** Graeme J. Ackland, Katherine Hampshire.

**Investigation:** Osman Adams, Simon Mariwah, Daniel Amoako-Sakyi, Fiifi Amoako Johnson, Katherine Hampshire.

**Methodology:** Graeme J. Ackland, Osman Adams, Katherine Hampshire.

**Project administration:** Osman Adams, Heather Hamill, Simon Mariwah, Fiifi Amoako Johnson, Katherine Hampshire.

**Resources:** Osman Adams, Heather Hamill, Katherine Hampshire.

**Software:** Graeme J. Ackland, Chia-Lin Wang.

**Supervision:** Graeme J. Ackland, Osman Adams, Simon Mariwah, Daniel Amoako-Sakyi, Fiifi Amoako Johnson, Katherine Hampshire.

**Validation:** Graeme J. Ackland, Osman Adams, Katherine Hampshire.

**Visualization:** Graeme J. Ackland, Katherine Hampshire.

**Writing – original draft:** Graeme J. Ackland, Osman Adams, Edmund Chattoe-Brown, Chia-Lin Wang.

**Writing – review & editing:** Graeme J. Ackland, Heather Hamill, Katherine Hampshire.

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
