## [Decision Letter · Decision Letter 0]

8 Jan 2026

Thank you for submitting your manuscript to PLOS ONE. After careful consideration, we feel that it has merit but does not fully meet PLOS ONE’s publication criteria as it currently stands. Therefore, we invite you to submit a revised version of the manuscript that addresses the points raised during the review process.

**ACADEMIC EDITOR:** Based on the publication criteria of PLOS ONE, we believe that the manuscript requiresimprovements to meet the expected standards of scientific quality and clarity. The reviewers’ comments are constructive and provide valuable guidance to enhance the robustness, transparency, and overall scientific merit of the study. Thus we strongly encourage the authors to carefully consider the reviewers’ suggestions and incorporate them where appropriate to strengthen the manuscript.regardsJose luiz vieira

We look forward to receiving your revised manuscript.

Kind regards,

José Luiz Fernandes Vieira

Academic Editor

PLOS One

Journal Requirements:

https://journals.plos.org/plosone/s/file?id=ba62/PLOSOne_formatting_sample_title_authors_affiliations.pdf....

“MRC funding the STREAMS collaboration with grant MR/T022132/1. OA GJA HH KH SM”

“We would like to thank the MRC for funding the STREAMS collaboration with grant MR/T022132/1.

“We note that you have provided additional information within the Acknowledgements Section that is not currently declared in your Funding Statement. Please note that funding information should not appear in the Acknowledgments section or other areas of your manuscript. We will only publish funding information present in the Funding Statement section of the online submission form.

“MRC funding the STREAMS collaboration with grant MR/T022132/1.  OA GJA HH KH SM”

7. Please ensure that you refer to Figure 5 in your text as, if accepted, production will need this reference to link the reader to the figure.

8. Please include a copy of Table 2 which you refer to in your text on page 12.

9. We note that Figures 1,2, 7-11 your submission contain [map/satellite] images which may be copyrighted. All PLOS content is published under the Creative Commons Attribution License (CC BY 4.0), which means that the manuscript, images, and Supporting Information files will be freely available online, and any third party is permitted to access, download, copy, distribute, and use these materials in any way, even commercially, with proper attribution. For these reasons, we cannot publish previously copyrighted maps or satellite images created using proprietary data, such as Google software (Google Maps, Street View, and Earth). For more information, see our copyright guidelines: http://journals.plos.org/plosone/s/licenses-and-copyright.

a. You may seek permission from the original copyright holder of Figures 1,2, 7-11 to publish the content specifically under the CC BY 4.0 license.

Please upload the completed Content Permission Form or other proof of granted permissions as an "Other" file with your submission

Reviewers' comments:

Reviewer's Responses to Questions

**Comments to the Author**

1. Is the manuscript technically sound, and do the data support the conclusions?

Reviewer #1: Partly

Reviewer #2: Yes

2. Has the statistical analysis been performed appropriately and rigorously?

Reviewer #1: Yes

Reviewer #2: Yes

3. Have the authors made all data underlying the findings in their manuscript fully available?

Reviewer #1: Yes

Reviewer #2: No

4. Is the manuscript presented in an intelligible fashion and written in standard English?

Reviewer #1: No

Reviewer #2: Yes

Reviewer #1: The manuscript presents an original and relevant empirical mapping of Ghana’s private antimalarial supply chain using network analysis. The study is clearly structured and ethically sound, and the use of directed network metrics adds value to the understanding of medicine distribution systems in LMICs. The manuscript is suitable for publication after essential revisions.

Major points that require attention include:

Sampling limitations must be more explicitly discussed, particularly regarding representativeness and potential bias in degree distribution, PageRank interpretation, and clustering metrics.

Node classification criteria should be clarified, especially for outlets with hybrid roles (manufacturer–wholesaler–retailer) and for handling self-loops.

Interpretation of PageRank should be more cautious, emphasizing that rankings reflect influence within the sampled sub-network and may not correspond to national-level dominance.

Degree distribution fitting requires additional methodological detail (sample size per distribution, treatment of zero-degree nodes, potential bias from unvisited suppliers).

Geographical analysis would benefit from quantitative support (e.g., distribution of geographic distances, correlations with path length).

Minor issues include improvements in figure readability, minor wording clarifications, and consistency of terminology.

Reviewer #2: 3. I kindly ask the authors to deposit the minimum set of de-identified data along with the analysis code in a public repository with a DOI and update the Data Availability Statement (DAS) accordingly. Once this has been completed, data availability will be in alignment with the PLOS ONE policy.

Reproducibility

Ensure that the deposited repository is fully reproducible by:

Including a single script or notebook capable of generating all reported statistics, figures, and tables from the manuscript without additional configuration.

DOI Citation

Update the Data Availability Statement (DAS) in the manuscript to include:

The DOI of the repository where the files are hosted.

The corresponding license for the data and code, ideally CC0 or CC BY.

Anonymization Note

Add a short paragraph under the Methods section

Describe the steps taken to de-identify or aggregate identifiers and locations.

Confirm that the public dataset is identical to the one used in the analysis, with only anonymization applied.

.

Reviewer #1: No

Reviewer #2: **Yes:** Tamyris Regina Matos LopesTamyris Regina Matos LopesTamyris Regina Matos LopesTamyris Regina Matos Lopes

---

## [Author Response · Author response to Decision Letter 1]

30 Jan 2026

Specific Editor Comments

1. We used the formatting from the latex template

% Template for PLoS

% Version 3.6 Aug 2022

2. We have used gitlab for code sharing

3. We added "The funders had no role in study design, data collection and analysis, decision to publish, or preparation of the manuscript." as requested

4. (a) There are no legal requirements

(b) The survey datasets are uploaded to the University of Edinburgh gitlab repostitory##

5. We moved the funding data from the Acknowledgements to the Funding Statement section.

6. We moved the ethics statement to the methods section

7&8. Fig 5 was erroneously labelled as Figure 12, from an old version, Sorry about thta . The undated Fig 5 contains all the information previously in Table 2, so that reference is removed.

9. The background maps in the figures use OpenStreetMap (OSM) data which is free for commercial use under the Open Database License (ODbL) 1.0. This is now properly acknowledged.

The details here are also provided in the cover letter to the editor

Reviewer 1

PONE-D-25-56182

An empirical network study of the antimalarial supply chain in Ghana

PLOS One

Reviewers' comments:

Reviewer's Responses to Questions

Comments to the Author

1. Is the manuscript technically sound, and do the data support the conclusions?

The manuscript must describe a technically sound piece of scientific research with data that supports

the conclusions. Experiments must have been conducted rigorously, with appropriate controls,

replication, and sample sizes. The conclusions must be drawn appropriately based on the data

presented.

Reviewer #1: Partly

Reviewer #2: Yes

2. Has the statistical analysis been performed appropriately and rigorously?

Reviewer #1: Yes

Reviewer #2: Yes

3. Have the authors made all data underlying the findings in their manuscript fully available?

The PLOS Data policy requires authors to make all data underlying the findings described in their

manuscript fully available without restriction, with rare exception (please refer to the Data Availability

Statement in the manuscript PDF file). The data should be provided as part of the manuscript or its

supporting information, or deposited to a public repository. For example, in addition to summary

statistics, the data points behind means, medians and variance measures should be available. If there

are restrictions on publicly sharing data—e.g. participant privacy or use of data from a third party—

those must be specified.

Reviewer #1: Yes

Reviewer #2: No

4. Is the manuscript presented in an intelligible fashion and written in standard English?

PLOS ONE does not copyedit accepted manuscripts, so the language in submitted articles must be clear,

correct, and unambiguous. Any typographical or grammatical errors should be corrected at revision, so

please note any specific errors here.

Reviewer #1: No

Reviewer #2: Yes

5. Review Comments to the Author

Please use the space provided to explain your answers to the questions above. You may also include

additional comments for the author, including concerns about dual publication, research ethics, or

publication ethics. (Please upload your review as an attachment if it exceeds 20,000 characters)

Reviewer #1: The manuscript presents an original and relevant empirical mapping of Ghana's private

antimalarial supply chain using network analysis. The study is clearly structured and ethically sound, and

the use of directed network metrics adds value to the understanding of medicine distribution systems in

LMICs. The manuscript is suitable for publication after essential revisions.

Major points that require attention include:

Sampling limitations must be more explicitly discussed, particularly regarding representativeness and

potential bias in degree distribution, PageRank interpretation, and clustering metrics.

We added a new section discussing the sampling challenges to the Methods.

Node classification criteria should be clarified, especially for outlets with hybrid roles (manufacturer

wholesaler retailer) and for handling self-loops.

The node classifications are fully defined in the Methods as follows:

Importer: out-links only

Wholesaler: in-links and out-links

Pharmacy/Wholesaler: in-links, out-links and customers.

Licensed Pharmacy/OTC: in-links and customers.

Interpretation of PageRank should be more cautious, emphasizing that rankings reflect influence within

the sampled sub-network and may not correspond to national-level dominance.

We address this specifically in the new Sampling Limitations section.

Degree distribution fitting requires additional methodological detail (sample size per distribution,

treatment of zero-degree nodes, potential bias from unvisited suppliers).

We add a note to the figure that the sample size is provided in the data spreadsheet, for which the DOI is given in the Data Availability Statement. The network is built by link-tracing, so there are no zero-degree nodes. Bias from unvisited suppliers is addressed in the new Sampling Limitations section.

Geographical analysis would benefit from quantitative support (e.g., distribution of geographic

distances, correlations with path length).

Geographic distances are shown schematically in the figure. We decided against attempting to make these quantitative as there is a huge variety in road quality, including seasonally, and we did not collect data about the distribution methods, so we cannot consider boats on Lake Volta, air and rail hubs. We now include in the online material a zoomable version of the background maps which enable the reader to quickly assess this information.

Minor issues include improvements in figure readability, minor wording clarifications, and consistency of terminology.

We used a professional proofreader, Kate Kilpatrick the improve the manuscript clarity (see Acknowledgements). We note that the background maps are not intended to be read in detail, but we have now made zoomable versions available online for the interested reader.

Reviewer 2

I kindly ask the authors to deposit the minimum set of de-identified data along with the

analysis code in a public repository with a DOI and update the Data Availability Statement (DAS)

accordingly. Once this has been completed, data availability will be in alignment with the PLOS ONE policy.

Reproducibility

Ensure that the deposited repository is fully reproducible by:

Including a single script or notebook capable of generating all reported statistics, figures, and tables

from the manuscript without additional configuration.

DOI Citation

Update the Data Availability Statement (DAS) in the manuscript to include:

The DOI of the repository where the files are hosted.

The corresponding license for the data and code, ideally CC0 or CC BY.

Anonymization Note

Add a short paragraph under the Methods section

Describe the steps taken to de-identify or aggregate identifiers and locations.

Confirm that the public dataset is identical to the one used in the analysis, with only anonymization

applied.

We have gathered/rewritten the analysis codes used by various team members into a single jupyter/python notebook and made this publically available as a gitlab repository with a CC-BY license.

This includes the survey data and no further anonymisation have been applied. In order to view very fine map details the reader is encouraged to use the zoomable maps using the pharmacies\_in\_ghana.html tool which links to OpenStreetMap. The survey data, zoomable maps and code used to generate data is available at

https://git.ecdf.ed.ac.uk/gja/ghananetworks

which is licensed under Creative Commons Attribution 4.0 International.

---

## [Editor Report · Decision Letter 1]

23 Mar 2026

An empirical network study of the antimalarial supply chain in Ghana

PONE-D-25-56182R1

Dear Dr. Ackland

We’re pleased to inform you that your manuscript has been judged scientifically suitable for publication and will be formally accepted for publication once it meets all outstanding technical requirements. All the reviewers suggestions were accepted by the authors.

Kind regards,

José Luiz Fernandes Vieira

Academic Editor

PLOS One

---

## [Editor Report · Acceptance letter]

PONE-D-25-56182R1

PLOS One

Dear Dr. Ackland,

I'm pleased to inform you that your manuscript has been deemed suitable for publication in PLOS One. Congratulations! Your manuscript is now being handed over to our production team.

Kind regards,

on behalf of

Dr. José Luiz Fernandes Vieira

Academic Editor

PLOS One